# Molecular Dynamics Simulation on Nanoindentation of M50 Bearing Steel

**DOI:** 10.3390/ma16062386

**Published:** 2023-03-16

**Authors:** Xuyang Hu, Lei Yang, Xunkai Wei, Hao Wang, Guoru Fu

**Affiliations:** 1Key Laboratory of Education Ministry for Modern Design and Rotor-Bearing System, Xi’an Jiaotong University, Xi’an 710049, China; 2Beijing Aeronautical Engineering Technical Research Center, Beijing 100076, China

**Keywords:** M50 bearing steel, molecular dynamics, nanoindentation, size effect

## Abstract

M50 bearing steel has great potential for applications in the field of aerospace engineering, as it exhibits outstanding mechanical and physical properties. From a microscopic point of view, bearing wear originates from the microscopic region of the contact interface, which usually only contains hundreds or even several atomic layers. However, the existing researches seldom study the wear of M50 bearing steel on the microscopic scale. This study explored the atomic-scale modeling method of M50 bearing steel. Then molecular dynamics simulations of nanoindentation on the M50 bearing steel model were carried out to study the size effect of the mechanical behaviors. The simulation results show that with the change in the radius of the diamond indenter in the nanoindentation simulation, the calculated nanohardness decreases. According to the size effect, when the indentation radius is 200 nm, the hardness obtained by the simulation is about 9.26 GPa, and that of the M50 sample measured by the nanoindentation is 10.4 GPa. Then nanoindentation simulations were carried out at different temperatures. The main bearings of aero-engines generally work at 300–500 degrees Celsius. When the simulated temperature was increased from 300 K to 800 K, the hardness of the model decreased by 15%, and the model was more prone to plastic deformation. In this study, a new molecular dynamics modeling method for M50 bearing steel was proposed, and then nanoindentation simulation was carried out, and the nanoindentation experiment verified the correctness of the model. These results are beneficial to the basic understanding of the mechanical performance of M50 bearing steel.

## 1. Introduction

As a rotating body supporting mechanical work, bearings are widely used in aerospace, ships, automobiles, and other major equipment, known as the “joint” of mechanical equipment [1,2]. M50 bearing steel, as a common aviation-bearing material, has become the main technical development direction [3,4,5,6,7]. The material has been transformed from the macroscopic continuum at the micro-scale into a discrete medium connected by atoms and molecules. However, the existing experimental methods are difficult to analyze the microscopic changes of atoms inside the damaged site, and the finite element analysis method based on continuum mechanics is no longer applicable at the microscopic scale. Therefore, it is necessary to conduct the molecular dynamics simulation for M50 bearing steel. Scratch and indentation is a contact loading process widely used to test various properties of materials, such as elastic modulus, hardness, fracture toughness, and surface adhesion [8,9,10]. The molecular dynamics simulation method can compensate for the defects of practical experimental methods by constructing microscale models and quantitatively reproducing the dynamic processes in real solids [8,9,10,11,12,13,14]. Therefore, it has become important for studying friction and wear mechanisms. In 2021, Hoang-Thien, Luu, et al. [15] conducted nanoindentation simulations using two different materials, aluminum and iron, studying different simulation parameters such as substrate size, indenter size, and loading rate. Furthermore, they analyzed how the properties of the substrate affect the critical forces needed to produce a pop-up event in the substrate. AlMotasem, A.T. et al. [16] used embedded atom method potential (EAM) to conduct extensive large-scale molecular dynamics simulations of nanoindentation/nanoscratch of nanocrystalline iron to explore the grain size dependence of wear response. The results show no obvious correlation between the friction and normal force and the grain size, and the single crystal (SC) iron has higher friction and normal force than the nanocrystalline(NC) sample.

By simulating nanoindentation experiments on the surface of single-crystal copper (100), Liu [17] explained the causes of a load drop in the load-depth curve and hardness drop in the hardness-depth curve during the plastic deformation of the sample and focused on the dislocation behavior in the sample under the indenter. In the indentation process, the ratio of the contact area between the indenter and the specimen, the ratio of the load to the projection of the contact area, namely the hardness in common sense, is obtained. The spatiotemporal distribution of the dislocation in the specimen around the indenter is calculated and analyzed. Wang [18] found both the size effect and grain boundary effect in the process of simulating gradient polycrystalline copper nanoindentation and used the N-G model to explain the size effect and grain boundary effect, and the simulation results were verified. Minor et al. [19] showed that there was a strong correlation between the occurrence of plasticity and dislocation activity during the nanoindentation process of aluminum crystals. Previous studies showed little work on molecular dynamics modeling of M50 bearing steel. Most of them used pure iron or iron-carbon alloy instead of various alloy steel and did not take into account the influence of various alloy elements on the results, which would lead to a large error.

Most studies of nanoindentation using molecular dynamics simulation focus on the mechanical response and deformation mechanism of the substrate. Generally, the hardness obtained at the nanoscale is larger than that in macroscopic experiments and varies with the diameter of the indenter [20,21,22,23,24,25]. However, it is not clear how the simulation results of the nanoindentation compare with the experimental data. 

Therefore, it is necessary to study the size effect of indenter radius on hardness in nanoindentation simulation to compare the actual hardness value of macroscopic materials, which may provide a useful reference for the results of nanoindentation experiments in the future. 

In this study, the multi-atomic molecular dynamics modeling of M50 bearing steel was carried out for the first time, and a method of comparing the experimental results with the simulation results was proposed to verify the correctness of the model. Because M50 bearing steel contains many elements, to make the established model conform to the mechanical properties of M50 bearing steel to the greatest extent, we select five elements (iron, carbon, chromium, molybdenum, and vanadium) to establish the molecular dynamics model of M50 bearing steel. Moreover, each element in the established model conforms to the requirements of the mass fraction of elements in M50 bearing steel. Nanoindentation simulation was performed on the M50 plate model, and the size effect in nanoindentation simulation was investigated. The influence of temperature on the hardness of the M50 plate model was studied. Nanoindentation experiments were carried out on M50 steel samples, and the molecular dynamics simulation results were verified. 

## 2. Computational Methods

### 2.1. Computational Model

The iron matrix model was established using the built-in commands of LAMMPS software (24 December 2020) [26]. The crystal direction [100], [010], and [001] were taken as the X, Y, and Z axes of the model, and the model size was set as 30$a_{0}$ × 30$a_{0}$ × 30$a_{0}$, $a_{0}$ is the lattice constant [27] of Fe. The lattice type was bcc.

The iron atoms in the iron matrix model were randomly replaced with a given number of chromium, molybdenum, and vanadium atoms by the method of atomic replacement, and the three atoms were evenly distributed in the iron matrix. A certain amount of carbon atoms were randomly inserted within the model range of the iron matrix by random insertion, which also ensures their uniform distribution. At this point, the mass fraction of each element in the model meets the mass fraction range of M50 bearing steel [28]. Table 1 [29,30] shows the chemical composition of the main elements of M50 bearing steel:

Then, the diamond hemisphere models with different radii were established at 5 Å above the surface of the M50 plate and set as rigid bodies, so there was no interaction between the diamond indenter and the M50 plate at the initial position. The established MD simulation model is shown in Figure 1.

Periodic boundary conditions are applied in the model’s X, Y, and Z directions. It can be seen that the model is divided into upper and lower parts. All the diamond indenters on the upper part are set as rigid bodies, and they do not deform. The orange part below is the fixed layer of the model, which does not have internal deformation and plays a role in supporting the model and bearing the load during the simulation. The blue atoms are called thermostatic layer atoms. The Nose-Hoover thermal baths [31] are applied on the atoms of the thermostatic layer so that they can provide a constant heat source for the atoms of the thermostatic layer, keeping the temperature of the system constant. The gray atoms are called Newtonian layer atoms. In the simulation, these atoms follow Newtonian mechanics, interact with other atoms, and also undergo deformation.

### 2.2. Potential Functions

According to the types of elements added, the interaction mode between the elements in the model is determined, and the potential function between each element in the model is set, selecting the type of potential function that best fits the model. The modified embedded atom method (MEAM) is used to simulate the interaction between Fe-C, Fe-Mo, Cr-Mo, and Mo-V, which can describe the properties of the alloy system well. In this study, the potential parameters between Fe and C were developed by B.-J. Lee et al. The potential parameters between Mo and V were developed by Wang, J et al. The potential parameters between Mo and Fe, Mo-Cr were developed by Sang-Ho Oh et al. [32,33,34].

The embedded atom method (EAM) was used to simulate the interaction potentials between Fe-Cr atoms and between Fe-V atoms. The EAM potential parameters used in the study were developed by Stukowski and Mendelev [35,36]. The interaction between other atoms is simulated by Lennard-Jones potential [37,38,39], and the potential parameters are all taken from the general force field. The potential parameters are given in Table 2.

### 2.3. Simulation Procedure

The simulations were all performed using the classical open-source MD LAMMPS code [40]. The simulation is divided into four parts: The first step is energy minimization: During model building since carbon atoms are randomly inserted into the matrix, carbon atoms can appear at any position in the matrix. As a result, they might coincide with other existing atoms or be too close to each other, which will lead to energy and stress explosion in the system. Therefore, it is necessary to simply optimize the system to eliminate the problems of atom coincidence and too small spacing caused by randomly inserted carbon atoms.

The second step is relaxation. The simulation of M50 bearing steel at high temperature melting, cooling, and aging treatment at room temperature can fully eliminate the structural defects and cracks in the model, reducing the residual stress in the model and making it closer to the physical and mechanical properties of the M50 bearing steel. The whole relaxation process was performed for 16 ps, including 5 ps in the high-temperature melting stage, 10 ps in the cooling stage, and 1 ps in the room-temperature aging stage. After this process, the energy of the system reached the lowest state, and the model also reached the most stable state.

The third step is the nanoindentation process. The diamond indenter was set as a rigid body and slowly pressed into the M50 bearing steel plate at a certain speed. The pressing depth at every moment and the load on the M50 plate were obtained. Then the nanoindentation hardness value of the M50 bearing steel was calculated. The second step applied the canonical ensemble (NVT) to both the thermostatic and Newtonian layers. In the third step, the canonical ensemble applied to the Newtonian layer was changed into a micro-canonical ensemble (NVE). This study used the velocity Verlet algorithm [41] to calculate the atomic motions. A time step of 0.0001 ps was used for the simulations.

In the nanoindentation simulation, load and penetration depth changes were tracked by monitoring the Z-direction force acting on the diamond tip and the distance the tip was pushed into the model. Fourth, the temperature of the thermostatic layer was changed, and the diamond indenter of the same radius was used respectively to simulate the nanoindentation process and to explore the influence rule of temperature on the hardness of the model.

The following formula was used to calculate the hardness in the nanoindentation simulation [42,43]:(1)$$H=\frac{P}{A}$$
where H is the model hardness value, *P* is the force on the diamond indenter at every moment, and *A* is the contact area between the diamond indenter and the base model at every corresponding moment. 

Figure 2 illustrates the calculation method of the model hardness during the nanoindentation process. 

The diamond indenter with radius *R* is uniformly pressed into the matrix, and the pressing depth is *d*. Because there is a cutoff radius between atoms, it must be considered to calculate the contact between the substrate and the diamond indenter. Therefore, at a pressing depth of *d*, the contact area of the diamond indenter on the M50 plate can be calculated as
(2)$$A=\pi {\;(\sqrt{R^{2}-{\left(R-d\right)}^{2}}+r_{c})}^{2}$$

Therefore, the nanoindentation hardness value of the model can be calculated by inserting it into the hardness calculation Formula (1). Figure 3 shows snapshots of the nanoindentation models with diamond indenters of different radius sizes.

### 2.4. Experimental Verification

To verify the accuracy of the fitting results based on the hardness data measured by indenters with different radius sizes in the nanoindentation simulation, we adopted the method of experimental verification. Firstly, the M50 bearing steel was cut into a suitable sample size of 10 mm × 10 mm × 2 mm. The sample surface is then polished on a high-speed polishing machine. Because the nanoindentation experiment is conducted on the nanoscale, the surface state of the sample has a particularly important impact on the test results [44,45,46], and the surface roughness also greatly impacts the discreteness of the test results [47,48,49]. So it is usually necessary to grind and polish the sample.

A Brooker Ti950 nanoindentation instrument with a berkvoich indenter with a radius of 200 nm was used to perform nanoindentation tests on samples [50,51,52]. A total of five nanoindentation tests were performed at five different points of the sample. During the test, the probe was pressed into the steel to a maximum depth of 140 nm, and the maximum load was 9000 nN, the loading time was 11.2 s, and the unloading time was 6 s.

## 3. Results and Discussion

### 3.1. Verification of Simulation Results

Unlike macroscopic experiments, there is a size effect in atomic scale simulation or experiment [53]. The size effect means that the measured physical quantity changes with the size of measuring tools, parameters during measurement, or the size of the measured sample. It is not an objective and constant quantity.

In the molecular dynamics simulation of aluminum and iron nanoindentation published by Hoang-Thien, Luu et al. [14] in 2021, it was found that when indenters of different sizes were pressed into the aluminum model, the measured hardness value of the aluminum base model decreased with the increase of the radius of the indenter, and finally remained roughly stable. In this study, we used diamond indenters with different radii to conduct nanoindentation simulation to explore the change rule of hardness measured by the diamond indenter.

Under the same periodic boundary conditions and temperature of 300 K, a diamond indenter with different radius sizes was pressed into the M50 bearing steel plate at a speed of 1 m/s, and the pressing depth was 12 Å. The Load-Distance curves of each nanoindentation test were obtained, as presented in Figure 4.

As can be seen from Figure 4, when the diamond indenter gradually presses down at a distance of about 2 Å, there is an adsorption phenomenon [20]. At this time, the indenter is subjected to a negative force in the direction of the Z axis, which is the adsorption force between the M50 plate and the diamond indenter. Moreover, it can be seen that the larger the radius of the diamond indenter is, the greater the adsorption force appears. 

We set the cutoff radius between the diamond indenter and the M50 plate as 1.5 Å; that is, when the pressing distance of the diamond indenter reaches 3.5 Å, it is considered that the diamond indenter and the M50 bearing steel model are just in contact.

As can be seen from Figure 4, the larger the radius of the diamond indenter, the greater the force on the diamond indenter when pressing down the same distance. To make the calculation more accurate, the stable force-displacement curve of the diamond indenter was selected, that is, the interval between 8–9 Å of the pressing distance, at which the indentation depth was 4.5–5.5 Å. The average load and displacement of the diamond indenter in this interval were substituted into the nanoindentation hardness calculation formula in Section 2. The average nanohardness of the diamond indenter with each radius in this interval was calculated respectively, and the results were plotted in Figure 5.

According to the hardness data of the model measured by the diamond nanoindentation simulation with each different radius, the data is fitted, and then the prediction is made. The predicted result is that when the radius of the indenter is 200 nm, the hardness measured by the model is about 9.26 GPa.

To verify the accuracy of the fitting results based on the hardness data measured by indenters with different radius sizes in the nanoindentation simulation, we adopted experiments to verify it. The load-distance curves of five nanoindentation tests are obtained by five repeated nanoindentation tests, as shown in Figure 6.

Through curve fitting and analysis, nanoindentation hardness values measured by each loading-unloading curve are obtained, which are 10.85 GPa, 10.14 GPa, 10.41 GPa, 10.62 GPa, and 9.98 GPa, respectively. The results are plotted in Figure 7.

It can be calculated from the data in Figure 7 that the average hardness of the results of five nanoindentation experiments is 10.4 GPa. Compared with the 9.26 GPa predicted by the 200 nm indenter during the simulation, the prediction error is very reasonable, only about 12%. Xiong et al. [54] also conducted a nanoindentation test on the M50 sample. The results show that the average nanohardness of the as-received M50 bearing steel is 9.03 GPa. As the radius of the indenter used in the nanoindentation test is generally several hundred nanometers, there is a certain size effect at this scale, so the measured data is affected by the radius of the indenters. Compared with the paper mentioned above, the error of hardness value of M50 steel is only 15%, which is reasonable.

Therefore, it can be verified that our fitting prediction of the simulation results is correct. Therefore, it can be concluded that the model of M50 bearing steel established in this study is close to the mechanical properties of real M50 bearing steel, and this model can be used as the molecular dynamics model of M50 bearing steel for subsequent research. After getting the correct model, a variety of simulation tests can be carried out on M50 bearing steel, such as simulation of adhesive wear, simulation plastic of deformation, and simulation of bearing changes under high temperature and high pressure, which are of guiding significance for bearing design. Therefore, our study now plays a basic role, and after the completion of this study, this model will be used for further research.

### 3.2. Influence of Temperature on the Hardness of the Model

Because the bearings of aero-engine work for a long time under high temperatures, high pressure, and extreme working conditions of oil, some mechanical properties at normal temperatures cannot explain the real working conditions [55]. Therefore, it is necessary to explore the mechanical properties of M50 under high temperatures [56], which can provide a reference for the hardness design of the bearing. We used a diamond indenter with a radius of 100 Å and a temperature range of 300 K–800 K to conduct nanoindentation simulation on the M50 bearing steel model. Then, the nanoindentation simulation of the loading and unloading process was carried out on the M50 bearing steel model at 300 K. The curves of the force and displacement acting on the diamond indenter were obtained, as shown in Figure 8.

In nanoindentation experiments, plastic deformation is usually determined by measuring the relationship between the load generated when the indenter applies pressure to the sample and the displacement between the indenter and the sample. This relationship can be obtained by recording the Load-Distance curve [57].

In the nanoindentation experiment, the commonly used method to determine whether plastic deformation occurs is based on two main characteristics in the load-distance curve: elastic stage and plastic stage. In the elastic phase, the relationship between load and distance is linear, which indicates that the sample is undergoing elastic deformation but not plastic deformation. When the load reaches a certain value, the distance increases significantly, indicating that the sample is undergoing plastic deformation. This point is often referred to as the “elastic-plastic transition point” or “Young’s modulus point” [58].

It can be seen from Figure 8a that after the indenter is pressed to a certain depth, there will be a bulge peak, and then the stress curve drops rapidly. This is because the model has plastic deformation at this time, and this point is the “elastic-plastic transition point.” From Figure 8b, it can be found that the indenter is pressed into the model at a depth of 10 Å. However, in the unloading process, the load becomes 0 nN after only 4 Å upward movement of the indenter, so it can be determined that plastic deformation occurs in the model. In the reference *Molecular dynamics simulation of single crystal Cu nanoindentation* [59], nanoindentation simulations were performed on single-crystal copper. The P-H curve and the atomic strain energy-depth curve were measured during the nanoindentation process, and the trend of the two curves was consistent. Therefore, the decrease in the atomic strain energy can be judged by the decrease in the load. Therefore, it can be concluded from Figure 8a that the higher the model’s temperature, the earlier the point of plastic deformation occurs, and the model was more prone to plastic deformation. The state diagram of the model at this point is shown in Figure 9.

In addition, it can be seen from Figure 8 that the higher the temperature, the smaller the force on the indenter at the same pressing depth, so the smaller the hardness of the model can be calculated. The model hardness values at each temperature were calculated separately and plotted in the following figure, as shown in Figure 10.

The hardness value of the model decreases gradually with the increase in temperature. When the simulated temperature was increased from 300 K to 800 K, the hardness of the model decreased by 15% from 49.42 GPa to 42 GPa. Moreover, higher temperature makes plastic deformation more likely to occur. This is because the higher the temperature of the model, the higher the energy in the system and the more unstable the model is. In the 300 K–800 K model, the higher the temperature, the more metal bonds between atoms will also absorb more energy in the environment, and the bonds will more easily reach the breaking point. Therefore, as the temperature increases, the hardness value of the model decreases, and plastic deformation is more likely to occur.

## 4. Conclusions

In this work, molecular dynamics was used to simulate the nanoindentation of M50 bearing steel, and the simulation results were verified. Through this study, the following conclusions can be drawn:(1)The molecular dynamics model of M50 bearing steel was established. Firstly, Lammps software was used to establish the Fe matrix model, and then a certain amount of Cr, Mo, and V atoms were added to it in the way of random replacement, and then a certain amount of C atoms was added to it in the way of random insertion so that it could meet the mass fraction requirements of each element in M50 bearing steel;(2)To verify the correctness of the model, a nanoindentation simulation was carried out. Due to the size effect of nanoindentation, the nanohardness obtained by using diamond indenters with different sizes was different, and the larger the radius of the diamond indenter, the smaller the hardness obtained. Then, the nanohardness data were fitted to predict that when the radius of the indenter was 200 nm, the hardness of the M50 plate measured by the nanoindentation was 9.26 GPa;(3)Five nanoindentation tests were carried out on M50 bearing steel samples, and the average hardness of the five tests was obtained to be 10.4 GPa. The error was reasonable compared with the simulation results, so the M50 plate model was verified to be correct and can be used for subsequent research;(4)The mechanical properties at different temperatures of the M50 plate were obtained. The results show that the hardness of the model decreases gradually with the increase in temperature. Furthermore, higher temperature makes plastic deformation more likely to occur.

## Figures and Tables

**Figure 1 materials-16-02386-f001:**
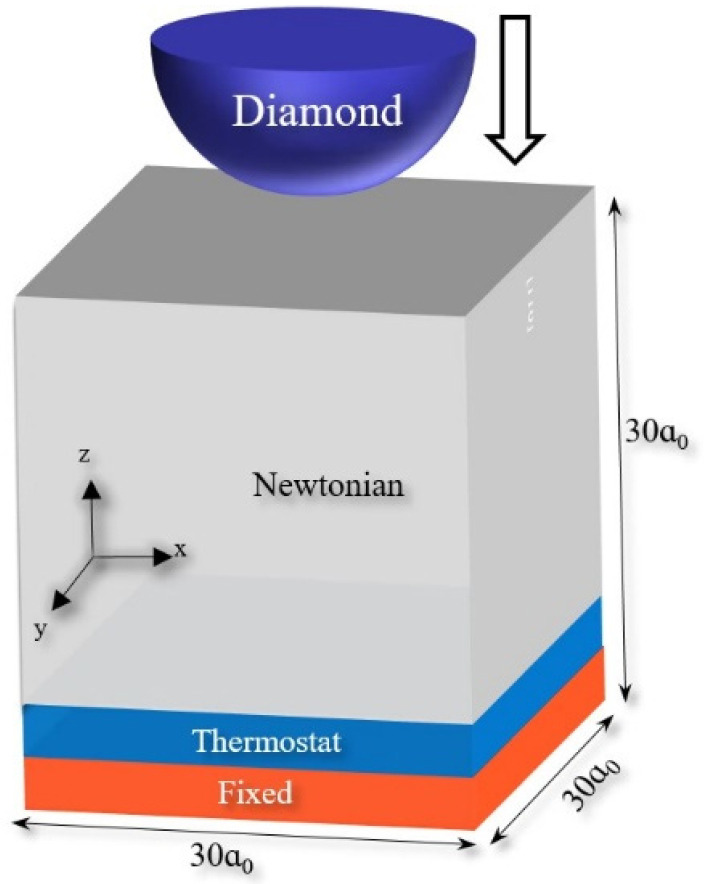
Schematic diagram of the MD simulation model: nanoindentation of M50 bearing steel. The orange fixed layer is 5$a_{0}$ thick, the blue thermostatic layer is 5$a_{0}$ thick, and the gray Newtonian layer has an atomic layer thickness of 20$a_{0}$. Here, the value of $a_{0}$ is 2.855 Å.

**Figure 2 materials-16-02386-f002:**
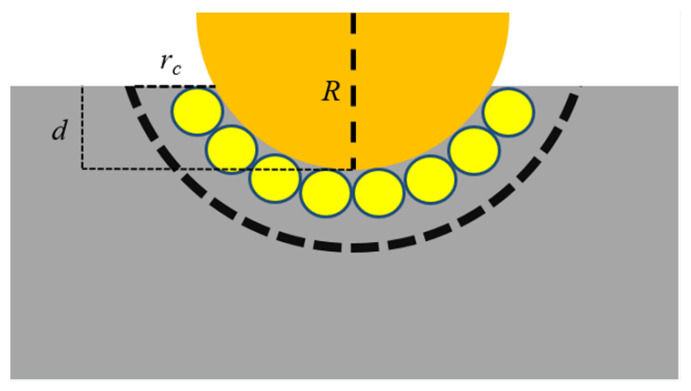
Schematic diagram of the hardness calculation method of the model when the indenter is pressed into the matrix at a depth of *d*.

**Figure 3 materials-16-02386-f003:**
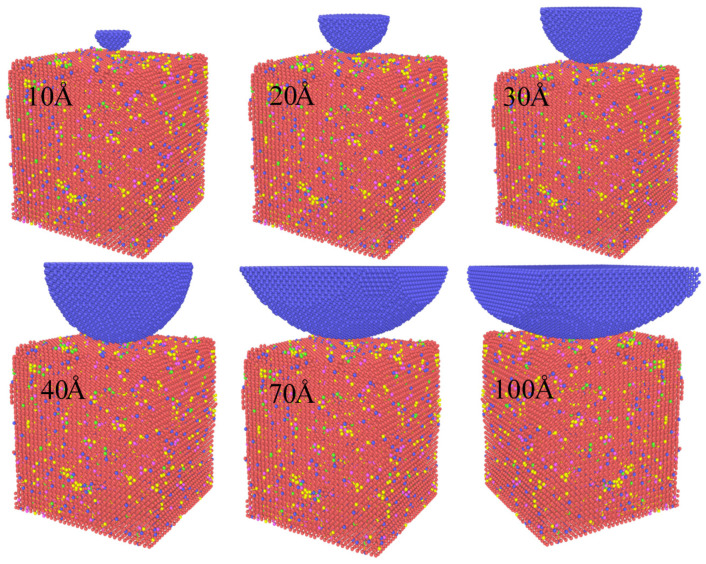
Snapshot of nanoindentation models for diamond indenters with different radius sizes. The radius of the diamond indenter is 10 Å, 20 Å, 30 Å, 40 Å, 70 Å, and 100 Å, respectively. The matrix model contains 56,790 Fe atoms, 2600 Cr atoms, 1500 Mo atoms, 630 V atoms, and 2200 C atoms.

**Figure 4 materials-16-02386-f004:**
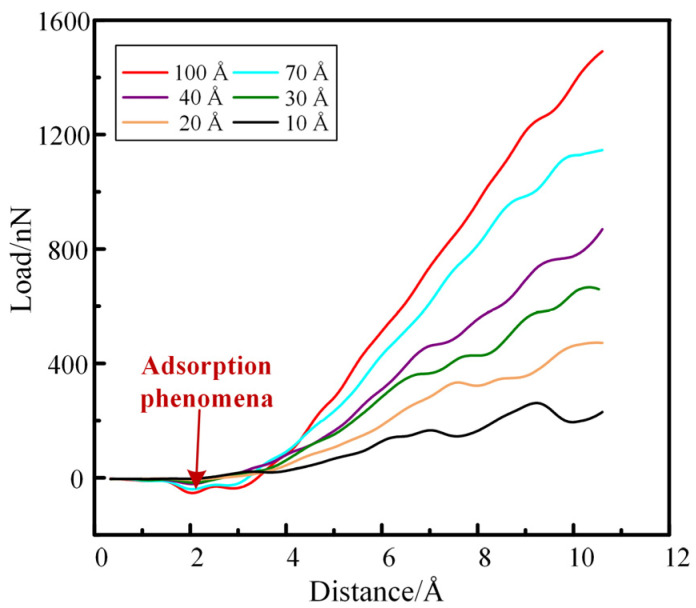
Load-Distance curve simulated by nanoindentation after smoothing treatment. 10 Å, 20 Å, 30 Å, 40 Å, 70 Å and 100 Å in the figure represent the radius of the diamond indenter in the simulation.

**Figure 5 materials-16-02386-f005:**
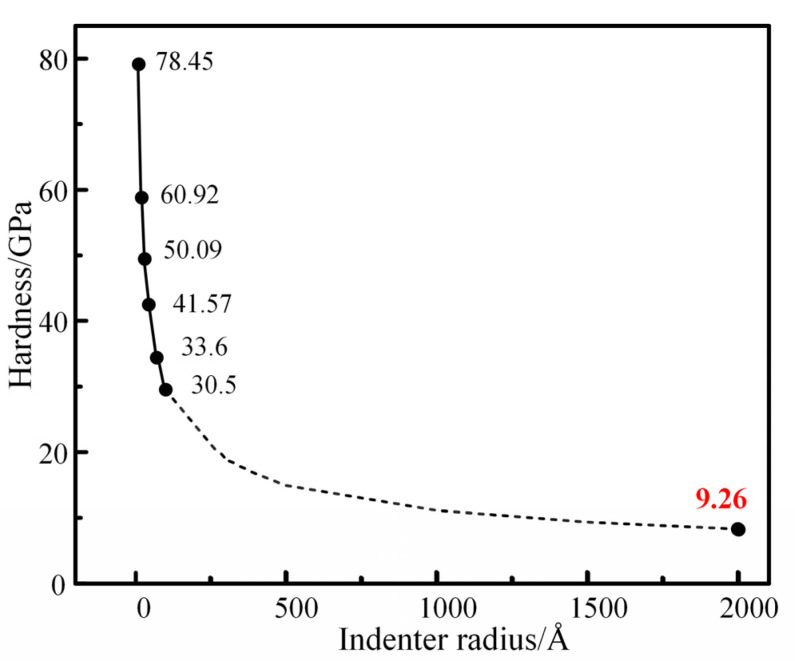
The fitting curves of matrix hardness values measured by nanoindentation simulation for diamond indenters with different radius sizes. It is predicted that when the radius of the diamond indenter is 200 nm, the hardness of the model measured is 9.26 GPa.

**Figure 6 materials-16-02386-f006:**
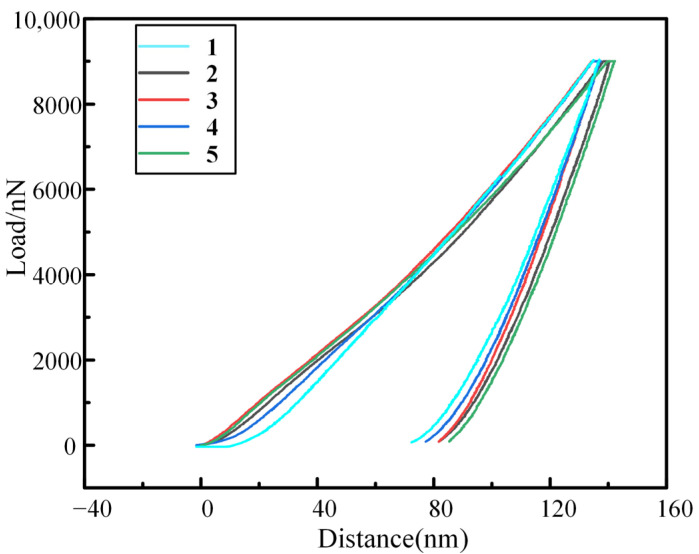
Five times loading-unloading cycle curve of M50 bearing steel in Nanoindentation test.

**Figure 7 materials-16-02386-f007:**
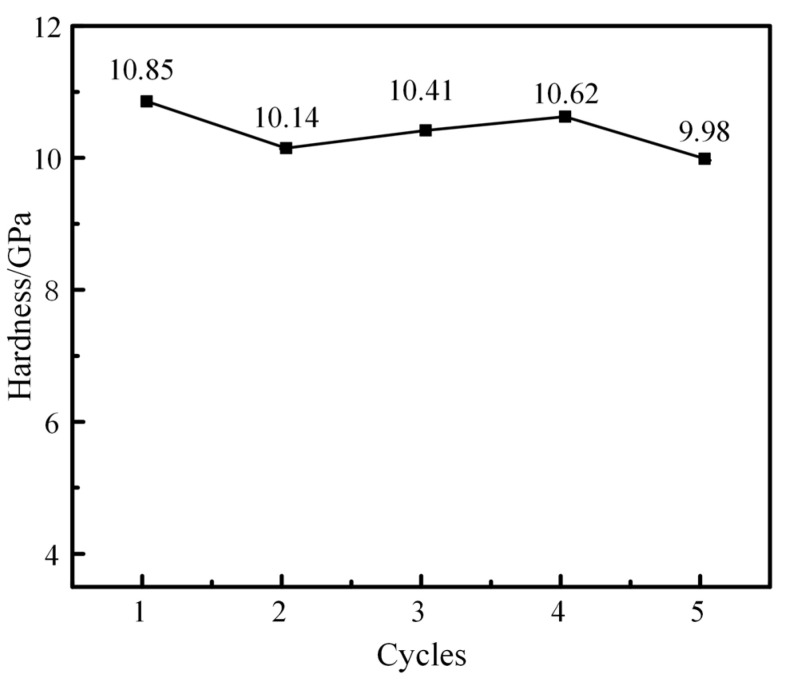
Hardness values of M50 bearing steel in five times nanoindentation test.

**Figure 8 materials-16-02386-f008:**
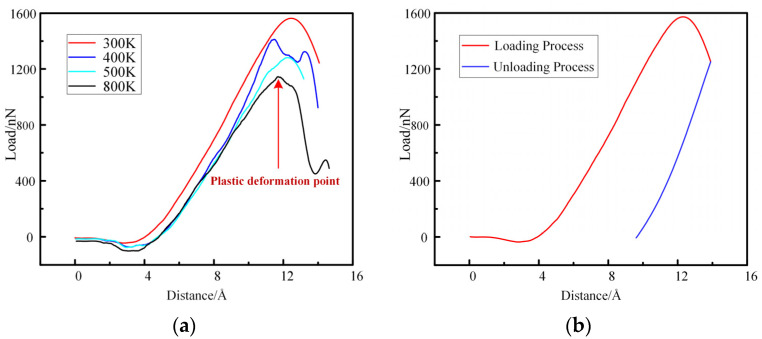
(**a**) Load-Distance curves of M50 bearing steel simulated by nanoindentation at different temperatures; (**b**) Load-Distance curve of M50 bearing steel simulated by nanoindentation during loading and unloading process at 300 K.

**Figure 9 materials-16-02386-f009:**
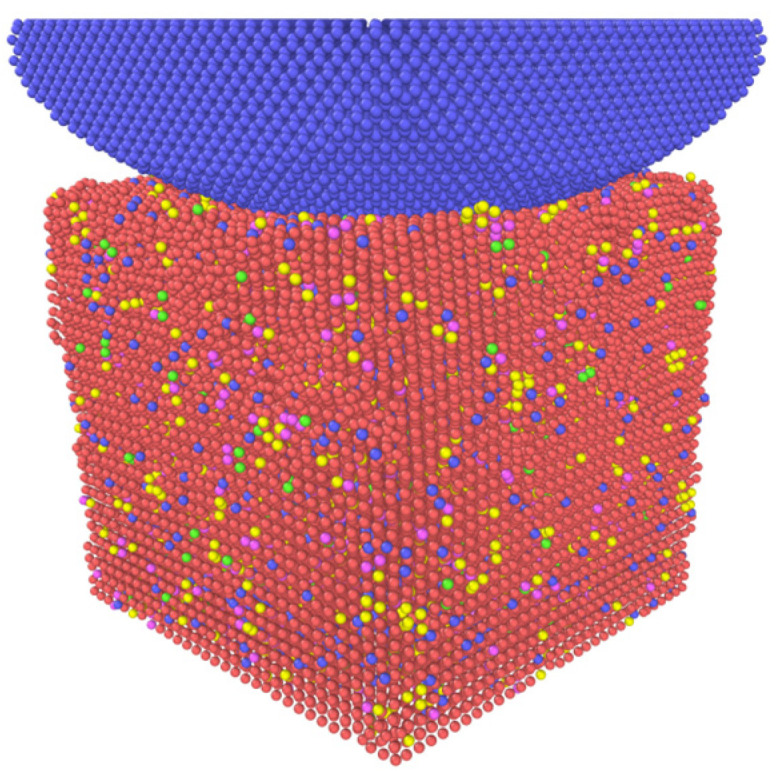
Snapshot of M50 bearing steel model when plastic deformation occurs.

**Figure 10 materials-16-02386-f010:**
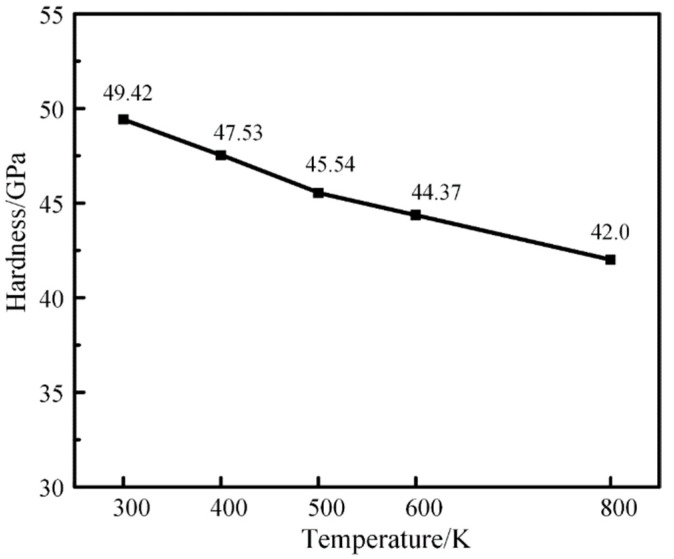
Nanoindentation hardness of M50 bearing steel model measured at different temperatures.

**Table 1 materials-16-02386-t001:** The chemical composition of the main elements of M50 bearing steel. (%, mass fraction).

Grade of Steel	C	Cr	Mo	V	Ni	S	P
M50	0.8–0.85	4–4.5	4–4.5	0.9–1.1	≤0.15	≤0.008	≤0.015

**Table 2 materials-16-02386-t002:** Lennard Jones energies (ε) and distances (σ) for Cr, V, Mo, and C.

Species	ε (eV)	σ (Å)
Cr-V	6.718 × 10^−4^	2.747
C-Cr	1.721 × 10^−3^	3.062
C-Mo	3.325 × 10^−3^	3.0749
C-V	1.777 × 10^−3^	3.1159

## Data Availability

Not applicable.

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
