# Peer review of "Molecular Dynamics Simulation on Nanoindentation of M50 Bearing Steel"

_materials, 2023, doi:10.3390/ma16062386_

Round 1

Reviewer 2 Report

The following comments can be made on the proposed paper.

1. The paper has a certain sense, but it is quite poorly written.

2. The English language should be improved, especially sentence constructions.

3. The title should be simpler and more adequate

4. Abstract...poorly written. As the temperature increases, the hardness decreases, this is not a discovery or a novelty.

5. It would be good if the work contained research on stress-deformation dependence, creep, fatigue, if possible.

6. Chapter 2. "The lattice is of bcc type...." it would be best seen from the dependence of Charpy impact energy versus temperature.

7. The chemical composition of the material is missing.

8. Table 1. Change labels, "epsilon" and "Sigma" are standard labels for strain and stress.

9. The experimentally determined hardness should be compared with the results known from the microhardness experiments of other researchers.

10. Fig. 7. GPa"

11. Figure 8. In order to claim plasticization, even local, one should have known "stress - strain" dependencies.

The work needs to be improved and supplemented in many segments.

Reviewer 3 Report

Reviewer # :  The authors reported the interesting results and conducted a significant work. The manuscript was well written and organized. However, there existed several issues that should be revised.

1.     Reformulate the abstract in order to clearly show the strengths of this work. Some other
qualitative must be added in the abstract.

2.     How does this work differ from the previously reported several works ??? Novelty of the study should be further highlighted.

3.     Why you did not use other more powerful techniques in characterization?

4.     Comparison with previous works are not reported.

5.     What is the social contribution of your research? How your results and approaches are useful for industrial sectors?

Thus, the manuscript should experience the major revision before acceptance.

Round 2

Reviewer 1 Report

Accept in the present form.

Author Response

Thank you very much.

Reviewer 2 Report

Although some answers were given, it is quite far from reality. Namely many answers are given mostly in a literature way. Take into account previously given remarks and try to make some better changes, The reviewer can not recognized any important improvement , even no new novelty or some importance. This related to chemical composition what is not defined  and is not directed to investigated material. Further, no  investigation is realized according to stress-strain relationship, nothing to fatigue or impact energy or another new investigation. However, no result is recognized as new one. Unfortunately, this paper is quite under the level of the journal. It is expected  that improvements need to be done n many areas. 

Reviewer 3 Report

I accept in this form

The authors respond to all my comments required.

Best regards

Author Response

Thank you very much.